# Effects of digital health counseling and behavioral interventions on weight management during pregnancy and postpartum: A systematic review and meta-analysis of randomized controlled trials

Sahar Khademioore[1]*, Alexandra M. Palumbo[1], Ahmad Sofi-Mahmudi[1,2], Taylor Incze[1], Kiara Pannozzo[1], Nicolette Christodoulakis[1], Rohan D'Souza[1,3], Gian Paolo Morgano[1], Nancy Santesso[1], Laura N. Anderson[1]

1 Department of Health Research Methods, Evidence, and Impact, McMaster University, Hamilton, Ontario, Canada, 2 National Pain Centre, Department of Anesthesia, McMaster University, Hamilton, Ontario, Canada, 3 Department of Obstetrics and Gynaecology, McMaster University, Hamilton, Ontario, Canada

* khades1@mcmaster.ca

## Abstract

### Objectives

This systematic review aimed to evaluate the effects of digital health counseling or behavioral weight management interventions for preventing excessive gestational weight gain (GWG) among pregnant individuals of all body mass index (BMI) categories, compared to routine care.

### Methods

We searched MEDLINE, Embase, CINAHL, ProQuest Dissertations/Theses, PsycINFO, and the Cochrane Central Register of Controlled Trials, up to February 2024. We included randomized controlled trials (RCTs) wherein pregnant women received counseling or behavioral interventions through digital health compared to routine care. Pairs of reviewers independently screened titles and abstracts and extracted data from eligible RCTs. Data were pooled using inverse-variance random-effects meta-analyses. We applied the Cochrane Risk of Bias 2.0 tool and the Grading of Recommendations Assessment, Development, and Evaluation (GRADE) approach to assess the magnitude and certainty of the effects.

### Results

We included 18 trials. Moderate certainty evidence showed 0.78 kg lower GWG in the weight management digital health intervention group, compared with routine care (95% CI: −1.40 to −0.16 kg). This reduction was higher in individuals with BMI ≥ 25 kg/

**Data availability statement:** The data used for all analyses, and analytic code used for all analyses are available on the OSF repository at https://doi.org/10.17605/OSF.IO/XRJU9.

**Funding:** The author(s) received no specific funding for this work.

**Competing interests:** The authors have declared that no competing interests exist.

m². Digital health interventions likely reduce the risk of excessive GWG (RR = 0.80; 95% CI: 0.68 to 0.95) and may result in little to no difference in the rate of cesarean birth (CB) (RR = 1.09; 95% CI: 0.81 to 1.48). Low-certainty evidence suggested that digital health weight management interventions may reduce the risk of gestational diabetes mellitus (GDM) (RR = 0.80; 95% CI: 0.57 to 1.12), pre-eclampsia (RR = 0.82; 95% CI: 0.51 to 1.33), and preterm birth (RR = 0.83; 95% CI: 0.53 to 1.28). High-certainty evidence showed that digital health weight management interventions have little to no effect on birthweight (MD = 0.00; 95% CI: −0.08 to 0.08).

## Conclusions

Digital health interventions are effective in reducing GWG and excessive GWG based on BMI. Additionally, evidence suggests that these interventions may lower the risk of GDM, pre-eclampsia, and preterm birth. However, their impact on birthweight, GWG across all BMI categories, and the risk of CB is trivial.

---

## Introduction

Gestational weight gain (GWG), defined as the change in weight during pregnancy, is calculated as the difference between pre-pregnancy weight and weight at childbirth [1]. The Institute of Medicine (IOM) has established recommended weight gain ranges based on pre-pregnancy body mass index (BMI), with gains exceeding these ranges classified as excessive GWG [2]. This condition is associated with increased risks of various adverse outcomes, including gestational diabetes mellitus (GDM), hypertensive disorders, and cesarean birth (CB) [3]. Furthermore, excessive GWG has been linked to preterm birth, small/large for gestational age infants, and childhood obesity [4]. Excessive GWG is highly prevalent. A systematic review of 1,309,136 participants from diverse international populations found that 47% of the individuals had excessive GWG, and 23% had inadequate GWG, as outlined by the IOM recommendations [4].

Traditional in-person weight management interventions have demonstrated effectiveness in moderately reducing mean GWG and associated maternal and child adverse outcomes [5]. These interventions typically employ multiple behavioral change techniques including dietary counseling, physical activity promotion, self-monitoring strategies, and educational components delivered through face-to-face sessions. However, these interventions often face significant resource and accessibility constraints. Digital health encompasses the use of information and communication technologies to improve healthcare delivery and patient outcomes [6]. This field includes mobile health applications, web-based platforms, telemedicine, and health informatics systems. Digital health interventions are increasingly used to support health behavior change and healthcare delivery, offering opportunities to reach individuals outside traditional clinical settings [7]. Studies have shown the effectiveness of digital health interventions in obesity prevention among the general population [8].

During the previous years, specifically during the COVID-19 pandemic (2020), there was an increase in the use of digital health interventions [9], and the results of these studies

need to be integrated into an updated evidence synthesis to make informed decisions about the development and implementation of digital health interventions in weight management during pregnancy. However, the latest systematic review related to this topic including only six studies reporting on GWG up to February 2019 [10]. Additionally, a comprehensive evaluation of digital health interventions should consider a broader range of maternal and child clinical outcomes beyond GWG, such as hypertension and diabetes, which were not fully addressed in this previous review.

Furthermore, since the presence of in-person elements in current studies may affect the true impact of digital health technologies by potentially masking their standalone effectiveness, it is necessary to distinguish between exclusively digital health interventions and those combining digital health with in-person components [11]. Therefore, there is a need for a comprehensive review focusing exclusively on digital health weight management interventions during pregnancy and postpartum that includes individuals of all BMI categories and investigate a broad range of maternal and child important outcomes. While the primary focus of prenatal weight management interventions is often on controlling excessive GWG, the ultimate goal extends beyond weight outcomes alone. This review aims to comprehensively evaluate both weight-related outcomes (including GWG and postpartum weight retention) and broader maternal and child health outcomes such as gestational diabetes, hypertensive disorders, and infant health measures.

## Objective

This systematic review aimed to evaluate the effects of digital health counseling or behavioral weight management interventions among pregnant individuals of all BMI categories, compared with routine care or in-person delivery of interventions.

## Methods

### Registration

We registered this systematic review with PROSPERO prospectively (CRD42023407325). Ethical approval was not required as only published data were used.

### Eligibility criteria

We included 1) randomized controlled trials (RCTs) that 2) enrolled individuals with a singleton pregnancy and aged 18 years or older; 3) behavioral interventions such as prescribed exercise and dietary programs focusing on weight management, that may also include a counseling component, and counseling only interventions focusing on weight management; 4) delivered the interventions over more than 12 weeks by a digital health technology either mobile, tablet or personal computers (including online platforms such as websites, text messages, emails, and applications); 5) compared to a group with either no digital health intervention, including routine pregnancy care or in-person interventions, We excluded 1) quasi-experimental or non-randomized trials that 2) enrolled individuals with pre-existing diabetes, mental health disorders, or were physically disabled, since GWG and other outcomes of the interest can be affected by these conditions and also they might not be able to perform exercise or adhere to the prescribed diet in the intervention, and 3) studies with in-person components other than the initial session.

For this review, "digital health" was defined as interventions delivered primarily through mobile devices, tablets, or computers, including mobile applications, text messaging, websites, email communications, and online platforms accessible via mobile devices. "Routine care" was defined as standard prenatal care without additional digital health components, which typically includes routine prenatal visits, standard weight monitoring, and general lifestyle advice as per local clinical guidelines.

### Data sources

We developed the search strategy in consultation with a medical librarian. On February 6, 2023, we conducted searches in MEDLINE, Embase, the Cumulative Index to Nursing and Allied Health Literature (CINAHL), PsycINFO, ProQuest

Dissertations/Theses, and the Cochrane Central Register of Controlled Trials (CENTRAL). The search strategy is available in S1 Appendix. There were no language restrictions. Additionally, we reviewed the reference lists of the studies we included and related reviews to identify any further studies that met our eligibility criteria. We updated the search on February 15, 2024.

### Study selection/screening

Using Rayyan (online systematic review software available at https://rayyan.ai), unfixed pairs of reviewers (SK, AP, TI, KP) working independently, screened titles and abstracts of all retrieved articles through our literature search and the full texts of all articles that met the criteria for potential eligibility. A third reviewer (LNA) resolved disagreements at each step.

### Data abstraction and risk of bias assessment

Reviewers worked independently in pairs to extract the following data into a piloted Excel spreadsheet: (1) study characteristics (i.e., setting, publication year, study design, country of origin), (2) patient characteristics (i.e., age, gestational age, pre-pregnancy weight, (3) details on the intervention and comparison (e.g., behavioral target, intervention delivery method, duration, frequency), and (4) outcomes including GWG, excessive GWG, GDM, pre-eclampsia, CB, postpartum weight retention, birthweight, and preterm birth.

Independently, pairs of reviewers performed the risk of bias assessment using the revised Cochrane risk-of-bias tool for randomized trials (RoB 2.0) [12]. Risk of bias was assessed as low, some concerns, or high, for the following domains: 1) randomization, 2) deviations from the intended interventions, 3) missing outcome data, 4) measurement of the outcome, and 5) selection of the reported result. Disagreement between reviewers was resolved through discussion or consultation with a third reviewer (LNA).

### Data synthesis and certainty in evidence

In studies with more than two arms, we selected the relevant intervention and comparison groups based on our eligibility criteria or combined all relevant intervention groups into a single group. We pooled the mean difference (MD) and its 95% confidence interval (CI) between groups for continuous outcomes using the inverse variance (IV) method to conduct meta-analyses. If studies provided pre-pregnancy and last weight before childbirth instead of GWG, we calculated GWG by subtracting the pre-pregnancy weight mean from the last weight before childbirth mean and reported it as the MD with the corresponding 95% CI. For dichotomous outcomes, we pooled risk ratios (RRs) and 95% CIs from each study using the Mantel–Haenszel method. We performed all the meta-analyses using random-effects models, since the results were likely to be influenced by clinical and methodological heterogeneity, using R statistical software [13] and the *meta* package [14].

### Subgroup and sensitivity analyses

Heterogeneity was determined by visual inspection of forest plots and $I^2$. We performed subgroup analyses only when high heterogeneity was detected, and when there were at least two trials in each subgroup: 1) BMI categories (BMI 18.5–24.9 vs. higher and lower BMI categories), 2) interactivity of the digital health intervention (interactive vs. non-interactive) defined as participants' interaction with the interventionists which could be a health care provider or a behavioral coach through digital health, 3) utilization of a smartphone application in the digital health intervention (application vs. no application (e.g., website)), and 4) overall risk of bias (high risk vs. low risk/some concerns). For all subgroup analyses, we tested for interaction using a Chi-square test [15].

## Certainty of evidence assessments

We assessed the certainty of the evidence as high, moderate, low, or very low for each pooled outcome using the Grading of Recommendations Assessment, Development and Evaluation (GRADE) approach [16]. We rated certainty according to the risk of bias of the studies, inconsistency, indirectness, publication bias, and imprecision. We assessed imprecision using a minimally contextualized approach, setting a null effect as the threshold for significance for all outcomes.

For outcomes with ≥10 studies contributing to the meta-analysis, we assessed publication bias using funnel plots and Egger's test [17]. We used the GRADEpro software (www.gradepro.org) to create the GRADE Summary of Findings tables showing absolute results for each outcome and reasons for the certainty of the evidence. We followed GRADE guidance for communicating our findings [18].

## Results

### Description of search results and studies

We identified 3,656 records and excluded 1,058 duplicates. After screening 2,598 titles and abstracts, 57 records were entered into the full-text stage, and 18 eligible RCTs (21 references) were identified (Fig 1). The main reason for exclusion was that trials were not exclusively digital health interventions (S1 Table). All studies were published in English, between 2014 and 2024.

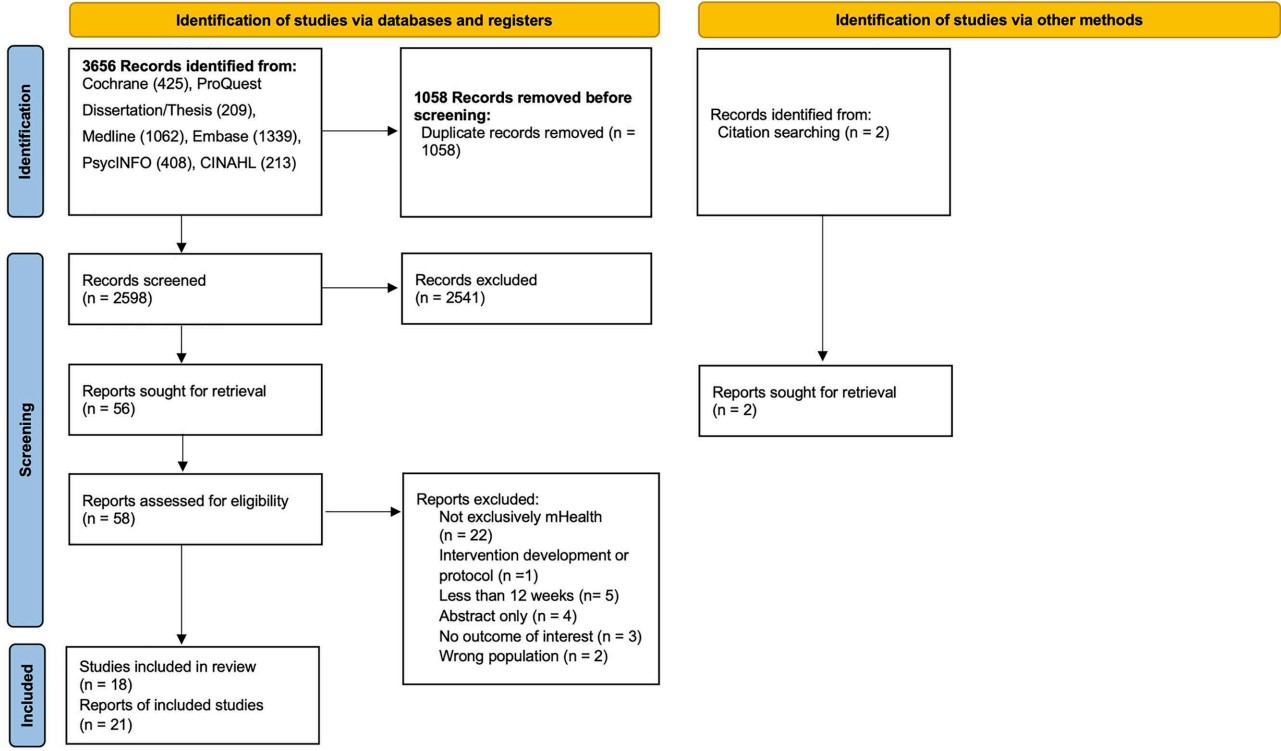

**Fig 1. PRISMA flow diagram for details of study selection.** *From:* Page MJ, McKenzie JE, Bossuyt PM, Boutron I, Hoffmann TC, Mulrow CD, et al. The PRISMA 2020 statement: an updated guideline for reporting systematic reviews. BMJ 2021;372:n71. https://doi.org/10.1136/bmj.n71.

## Study characteristics

Table 1 provides a summary of the characteristics of the included studies. Most studies were conducted in the United States (n = 8) [19–28], followed by Spain (n = 2) [29,30], with single studies each from Taiwan [31], China [32], Ireland [33], Japan [34], India [35], Sweden [36], Germany [37], and Australia [38]. Seven RCTs were randomized pilot or feasibility studies. Four RCTs spanned from pregnancy through the postpartum period. The total sample size varied from 26 to 1,689. Eight RCTs focused only on pregnant individuals with BMI ≥ 25 kg/m² [19,22,27,28,30–32,38] while one RCT included only women with a healthy BMI [34]. The remaining RCTs included women from all BMI categories [20,23–26,29,35–37]. All RCTs except one [33] targeted both healthy eating and physical activity; the remaining one focused exclusively on healthy eating. The digital health delivery methods varied across RCTs with three using text messaging only [26,34,35], and the remaining studies using a mix of digital health delivery approaches, such as mobile applications and websites. Only one study incorporated an in-person intervention group. Due to the lack of comparable data across studies, we limited our analysis to the standard care and digital health groups from this study. Therefore, we were unable to provide a comparison between digital health and in-person delivery in our study.

In terms of risk of bias, the measurement of the outcome was the domain with most at risk of bias often due to self-reported outcomes or unblinded outcome assessors. GWG in the included studies was measured by either self-report, trained assessors, or medical chart abstraction, and was calculated as the difference between final pregnancy weight (measured between 34–40 weeks gestation) and either pre-pregnancy weight or early pregnancy weight measured at 10–16 weeks gestation. Selective reporting of the outcome was the domain least at risk of bias. The risk of bias plots for each outcome are available in Figures A-G in S2 Appendix.

**Gestational weight gain (GWG).** Seventeen trials [19–21,23–32,34–39], including 3,231 participants, reported GWG. Moderate certainty evidence suggested 0.78 kg lower GWG in the weight management digital health intervention group, compared with routine care (95% CI: −1.40 kg to −0.16 kg; Tables 2 and 4 and Figure A in S3 Appendix). We performed a subgroup analysis due to high heterogeneity (I² = 47.2%). The subgroup analysis based on BMI categories explained the heterogeneity, showing that in individuals with BMI ≥ 25 (overweight or obesity), weight management digital health interventions may reduce GWG by 1.80 kg (95% CI: −2.60 kg to −0.99 kg, moderate certainty; test of interaction $P < 0.001$; Table 2 and Figure B in S3 Appendix), compared with routine care. However, in studies including all BMI categories there may be no effect (MD = −0.00; 95% CI: −0.05 kg to 0.04 kg; high certainty). There was no evidence of a statistically significant interaction in subgroup analyses by the risk of bias and intervention type (interactive vs. non-interactive and mobile application vs. other delivery methods) did not explain the heterogeneity (interaction tests p > 0.05; Table 2 and Figures C-E in S3 Appendix). Although these interactions were not statistically significant and confidence intervals overlapped, pooled effect estimates suggest that both interactive interventions (−1.51 kg; 95% CI: −2.71 to −0.32 vs. −0.48 kg; 95% CI: −1.14 to 0.19) and those using mobile applications (−1.06 kg; 95% CI: −1.85 to −0.27 vs. −0.55 kg; 95% CI: −1.54 to 0.44) may reduce GWG more effectively than non-interactive interventions and non-mobile application interventions.

**Gestational weight gain (GWG) exceeding Institute of Medicine (IOM) recommendations.** Fourteen trials [19–21,23–26,28–31,35–39], including 1,939 participants, provided data on the number of pregnancies with excessive GWG according to IOM guidelines. We found a RR of 0.80 (95% CI: 0.68 to 0.95; Table 3 and Figure F in S3 Appendix; moderate certainty) with an absolute effect of 9% fewer pregnant individuals with GWG exceeding the recommendations (from 15% to 2% fewer) when receiving digital health interventions (Table 4). Subgroup analysis by interactivity of the digital health intervention explained the heterogeneity (test of interaction p < 0.01; Figure H in S3 Appendix and Tables 3). Interactive digital health interventions may reduce the risk of excessive GWG by 41% (95% CI: 0.47 to 0.73) compared with routine care. However, no such effect was found in the non-interactive digital health interventions (RR 1.00; 95% CI: 0.88 to 1.13). Subgroup analyses by the risk of bias of studies, BMI, and the use of an application in the intervention did not explain the differences in effects (Table 3, and Figure G, Figure I and Figure J in S3 Appendix).

**Table 1. Characteristics of included studies.**

| Author | Country/ Setting | Mean age in years (SD) | Population | Pre-pregnancy BMI mean (SD) | Sample size | Type of trial | Behavioral targeted | Intervention delivery method/ name | Intervention details | Comparator |
|---|---|---|---|---|---|---|---|---|---|---|
| Chen 2022 | Taiwan/ Prenatal clinic | I: 31.56 (2.17) C: 32.65 (4.22) | Antenatal < 17 weeks, BMI ≥ 25 kg/m² | NR | 92 (I: 46, C: 46) | Pilot RCT (2 arms) | GWG, diet, PA | MyHealthyWeight app, activity tracker | mHealth app for GWG management, Mi Band 5 for 8,500 steps/day goal | Standard care |
| Coughlin 2020 | USA/ Obstetric practice | I: 32.7 (4.3) C: 30.6 (2.5) | Antenatal 11–16 weeks, BMI ≥ 18.5 kg/m² | NR | 26 (I: 13, C: 13) | Feasibility RCT | GWG, diet, PA | H42/H4U app (Lose It!) | Self-monitoring with COACH framework, learning activities, coach calls | A single, in-person health educa-tion session |
| Dahl 2018 | USA/ Social networking websites across USA | I: 30.2 (3.6) C: 29.9 (3.1) | Antenatal ≤ 20 weeks, BMI > 18.5 kg/m² | I: 26.1 (5.3) C: 27.0 (5.5) | 140 (I: 77, C: 63) | Parallel RCT (2 arms) | Diet, PA | MakeMe™ app and website | Healthy behavior challenges via app and website | Stress reduc-tion and man-agement app (MakeMe™) |
| Feng 2023 | China/ An obstetrics and gynecology hospital | I: 32.0 (15.25) C: 33.0 (14.98) | Antenatal 6–7 weeks, BMI ≥ 24 kg/m², primigravid | I: 28.9 (9.85) C: 29.34 (10.16) | 281 (I: 143, C: 138) | Parallel RCT (2 arms) | Weight, diet, PA | App + usual care | Personalized weight goals, calorie intake counting, daily tracking, reminders, educational content | Standard care |
| Gonzalez-Plaza 2022 | Spain/ Maternal–fetal department of a hospital clinic | I: 32.4 (5.4) C: 33.4 (4.7) | Antenatal 12–18 weeks, BMI ≥ 30 | I: 33.1 (2.9) C: 32.7 (3.3) | 150 (I: 78, C: 72) | Parallel RCT (2 arms) | GWG, PA | Smart band (Mi Band 2 and Mi Fit; Hangouts App) and app with mid-wife counseling | Inactivity alerts, goal rewards, SMS, videos via Hangouts app, midwife counseling | Standard care |
| Herring 2017 | USA/ Outpatient obstetric offices | I: 25.9 (4.9) C: 25.0 (5.7) | Antenatal < 20 weeks, BMI 25–45 kg/m² | NR | 66 (I: 33, C: 33) | Pilot RCT (2 arms) | GWG, diet, PA, postpar-tum weight loss | Text messages, Facebook posts linked to websites, calls | Daily skill-building texts, weekly Face-book posts, weekly to monthly calls for sup-port and self-efficacy | Standard care |
| Holmes 2020 | USA/ Prenatal clinics | I: 26.0 (5.4) C: 27.2 (5.51) | Antenatal 10–20 weeks, BMI 20–45 | I: 30.4 (6.04) C: 29.8 (5.42) | 83 (I: 42, C: 41) | Parallel RCT (2 arms) | Diet, PA | EX Texting (text messages) | SMS messages focused on energy intake and physical activity | General health SMS (no nutrition/ PA) |
| Kennedy 2018 | Ireland/ Antena-tal clinic in a uni-versity hospital | I: 31.48 (4.79) C: 31.92 (4.44) | Antenatal < 18 weeks | NR | 250 (I: 125, C: 125) | Parallel RCT (2 arms) | Diet | OptiMUM Nutrition (website, leaflets) | Recipes, nutrition, and lifestyle advice through a website | Standard care, healthy eating leaflets |
| Kodama 2021 | Japan/ Obstetric outpatient clinic | I: 30.7 (3.2) C: 31.5 (3.9) | Primigravid Antenatal ≤ 12 weeks, BMI 18.5–24.9 | I: 20.1 (1.8) C: 21.5 (3.5) | 28 (I: 15, C: 13) | Pilot RCT (2 arms) | Sleep, mental health, breastfeeding, weight, diet | Text messages | Pregnancy-related information on social services, breastfeed-ing, sleep, mental health, weight man-agement, and suitable meals | Standard care |
| Olson 2018 | USA/ Prenatal clinic, private obstetric prac-tices, ultrasound offices | Range: I: 31.8–35.7 C: 29.7–33.9 | Antenatal ≤ 20 weeks, BMI 18.5–34 | NR | 1689 (I: 1126, C: 563) | Parallel RCT (2 arms) | GWG, diet, PA | Website | Online/mobile behav-ioral intervention with tools for tracking weight gain, diet, PA, and health information | Standard care (no diet/PA tools) |
| Pollak 2014 | USA/ Prenatal clinics | I: 29 (5) C: 32 (2) | Antenatal 12–21 weeks, BMI 25–40 | I: 29 (4) C: 28 (5) | 33 (I: 22, C: 11) | Pilot RCT (2 arms) | GWG, diet, PA | PregCHAT (Tai-lored SMS) | Text messages to increase self-efficacy, improve outcome expectations, address barriers, promote self-monitoring | Txt4Baby (generic texting) |

*(Continued)*

**Table 1.** (Continued)

| Author | Country/ Setting | Mean age in years (SD) | Population | Pre-pregnancy BMI mean (SD) | Sample size | Type of trial | Behavioral targeted | Intervention delivery method/ name | Intervention details | Comparator |
|---|---|---|---|---|---|---|---|---|---|---|
| Rani 2022 | India/ Maternity hospital | I1: 26.66 (2.22) I2: 25.53 (2.67) I3: 25.77 (3.06) I4: 26.57 (2.64) C: 26.14 (2.80) | Antenatal < 16 weeks, BMI ≥ 18.5–30 | 22.40 (2.05) | 150 (I1: 30, I2: 30, I3: 30, I4: 30, C: 30) | Parallel RCT (5 arms) | Diet, PA | Text messages | 42 messages addressing well-being, myths, and tips, prepared with a gynecologist | Minimal care (healthy life-style advice) |
| Redman 2017 | USA/ Prenatal clinics | In-person: 29.2 (4.8) Remote: 29.0 (4.2) C: 29.5 (5.1) | First trimes-ter, BMI 25.0–39.9 kg/ m² | NR | 54 (I1: 19, I2: 18, C: 17) | Pilot RCT (3 arms) | GWG, diet, PA | SmartMoms (mobile phone, counseling) | Personalized GWG weight graph, daily self-monitoring of weight, diet, PA, behavioral modifica-tion tools | Standard care |
| Sandborg 2021 | Sweden/ Mater-nity clinics | I: 31.4 (4.3) C: 31.3 (3.8) | Antenatal early pregnancy (mean 13.9 weeks) | NA | 305 (I: 152, C: 153) | Parallel RCT (2 arms) | GWG, diet, PA | HealthyMoms (app) | App with informational themes, notifications, self-monitoring with feedback, recipes, exercise guide, videos, pregnancy calendar | Standard care |
| Smith 2016 | USA/ Prenatal clinics | I: 29.7 (4.1) C: 29.4 (4.9) | Antenatal 10–14 weeks, sedentary women | I: 27.3 (4.6) C: 25.4 (4.5) | 51 (I: 26, C: 25) | Parallel RCT (2 arms) | PA, diet | The Blossom project (website, forum, email, device, journal, calendar) | Increase PA to > 150 minutes of moderate PA/week (> 10-minute bouts) | Minimal care (exercise/diet tips) |
| Téoule 2024 | Germany/ University Medical Centre Mannheim | I: 32 (4) C: 32 (4) | < 20 weeks antenatal | I: 25 (5) C: 25 (4) | 104 (I: 52, C: 52) | Parallel RCT (2 arms) | GWG, diet, PA | Buddy Healthcare app, fitness tracker | Support from mid-wives/assistants via virtual health-coaching, Buddy Healthcare app for communication, educational materials, PA tracking | Fitness tracker, dif-ferent version of Buddy Healthcare app without pregnancy-related information, without the possibility to use the chat function |
| Uria-Minguito 2023 | Spain/ Hospital | I: 33.80 (3.27) C: 33.29 (5.27) | Antenatal (first prenatal visit) | I: 22.70 (4.17) C: 25.09 (5.40) | 260 (I: 130, C: 130) | Parallel RCT (2 arms) | GWG, PA | Online exercise program (Zoom, YouTube) | Supervised online moderate exercise program, three days a week, tailored to each trimester | Standard care |
| Willcox 2017 | Australia/ Ter-tiary hospital | I: 33.0 (3.4) C: 32.0 (5.1) | Antenatal 10–17 weeks, BMI > 25 | NR | 100 (I: 45, C: 46) | Pilot RCT (2 arms) | Diet, PA, GWG | SMS, video mes-sages, website, Facebook chat | Promoting positive health behaviors, monitoring goals, and self-monitoring of GWG | Standard care |

I: Intervention group; C: Control group; RCT: Randomized Controlled Trial; GWG: Gestational Weight Gain; PA: Physical Activity; SD: Standard Devia-tion; NR: Not Reported; BMI: Body Mass Index; USA: United States of America

**Table 2. Results of the meta-analysis and subgroup analysis of randomized trials investigating mHealth during pregnancy – Continuous outcomes.**

| Outcome/subgroup | | # of trials | MD (95% CI) | N analyzed | | I² | P value for test of interaction |
|---|---|---|---|---|---|---|---|
| | | | | mHealth | Con-trol | | |
| **Gestational weight gain (GWG)** | BMI ≥ 25 | 8 | −1.80 (−2.60, −0.99) | 360 | 352 | 0.0 | **≤0.001** |
| | All BMI | 8 | −0.00 (−0.05, 0.04) | 1525 | 930 | 0.0 | |
| | 18.5 < BMI < 24.9 | 2 | −0.43 (−2.97, 2.11) | 39 | 31 | 0.0 | |
| | Interactive | 6 | −1.51 (−2.71, −0.32) | 264 | 257 | 18.9 | 0.138 |
| | Non-interactive | 11 | −0.48 (−1.14, 0.19) | 1660 | 1056 | 38.7 | |
| | Used mobile app | 8 | −1.06 (−1.85, −0.27) | 476 | 468 | 7.8 | 0.297 |
| | No mobile app | 9 | −0.55 (−1.54, 0.44) | 1448 | 845 | 42.0 | |
| | High overall risk of bias | 2 | −0.41 (−3.82, 4.65) | 53 | 47 | 0.0 | 0.591 |
| | Low/some concern overall risk of bias | 15 | −0.82 (−1.45, −0.18) | 1871 | 1266 | 53.2 | |
| | *Total* | **17** | **−0.78 (−1.40, −0.16)** | **1924** | **1313** | **47.2** | **–** |
| **Birthweight** | BMI ≥ 25 | 2 | −0.22 (−0.37, −0.07) | 64 | 72 | 0.0 | **0.015** |
| | All BMI | 7 | 0.05 (−0.02, 0.11) | 1337 | 865 | 79.6 | |
| | 18.5 < BMI < 24.9 | 1 | −0.02 (−0.29, 0.26) | 15 | 13 | – | |
| | High overall risk of bias | 1 | −0.02 (−0.29, 0.26) | 15 | 13 | – | 0.941 |
| | Low/some concern overall risk of bias | 9 | −0.00 (−0.09, 0.08) | 1401 | 937 | 71 | |
| | *Total* | **9** | **−0.00 (−0.08, 0.08)** | **1416** | **950** | **66** | **–** |

CI: confidence interval; BMI: body mass index; MD: mean difference.

**Postpartum weight retention.** Four trials [21,23–25,35] provided data on postpartum weight retention. However, since the follow-up time varied from 1 to 12 months after childbirth, we did not conduct a meta-analysis on this outcome. Table 5 provides information regarding postpartum weight retention. Overall, except for one study [24,25] (at 4 weeks follow-up), the difference between postpartum weight and pre/early pregnancy weight was lower in the weight management digital health intervention group than in the routine care at different time points.

**Gestational diabetes mellitus (GDM).** Seven trials [20,22,29,30,32,33,37] (2,262 participants) provided data on the number of women with GDM. We found a RR of 0.80 (95% CI: 0.57 to 1.12; Table 3 and Figure K in S3 Appendix; low certainty) with an absolute effect of 3% fewer GDM cases (from 6 fewer to 2 more) in individuals receiving digital health interventions compared with routine care (Table 4). Analyses by the risk of bias of studies did not explain the differences in effects (Table 3 and Figure L in S3 Appendix).

**Table 3. Results of the meta-analysis and subgroup analysis of randomized trials investigating digital health during pregnancy – Dichotomous outcomes.**

| Outcome/subgroup | | # of trials | RR (95% CI) | N analyzed | | I² | P value for test of interaction |
|---|---|---|---|---|---|---|---|
| | | | | digital health | Control | | |
| Gestational weight gain (GWG) exceeding Institute of Medicine (IOM) recommendations | BMI ≥ 25 | 6 | 0.69 (0.56, 0.84) | 211 | 206 | 0.0 | 0.156 |
| | All BMI | 8 | 0.87 (0.67, 1.14) | 918 | 562 | 60.2 | |
| | 18.5 < BMI < 24.9 | 1 | 0.83 (0.43, 1.61) | 10 | 9 | – | |
| | Interactive | 6 | 0.59 (0.47, 073) | 264 | 253 | 0.0 | **<0.001** |
| | Non-interactive | 8 | 1.00 (0.88, 1.13) | 889 | 533 | 3.9 | |
| | Used mobile app | 7 | 0.76 (0.62, 0.93) | 341 | 331 | 0.0 | 0.353 |
| | No mobile app | 7 | 0.85 (0.63, 1.14) | 812 | 455 | 67.4 | |
| | High overall risk of bias | 1 | 1.21 (0.79, 1.85) | 38 | 34 | – | 0.165 |
| | Low/some concerns overall risk of bias | 13 | 0.78 (0.65, 0.93) | 1115 | 752 | 51.7 | |
| | *Total* | **14** | **0.80 (0.68, 0.95)** | **1153** | **798** | **52.1** | **–** |
| Gestational diabetes mellitus (GDM) | BMI ≥ 25 | 3 | 0.93 (0.70, 1.22) | 227 | 217 | 0.0 | 0.561 |
| | All BMI | 4 | 0.66 (0.33, 1.34) | 1130 | 688 | 58.7 | |
| | 18.5 < BMI < 24.9 | 0 | – | – | – | – | |
| | High overall risk of bias | 0 | – | – | – | – | – |
| | Low/some concerns overall risk of bias | 7 | 0.80 (0.57, 1.12) | 1357 | 905 | 30.1 | |
| | *Total* | **7** | **0.80 (0.57, 1.12)** | **1357** | **905** | **30.1** | **–** |
| Cesarean birth (CB) | BMI ≥ 25 | 4 | 1.19 (0.75, 1.86) | 343 | 345 | 68.1 | 0.612 |
| | All BMI | 4 | 0.99 (0.57, 1.71) | 1141 | 699 | 66.5 | |
| | 18.5 < BMI < 24.9 | 0 | – | – | – | – | |
| | Interactive | 4 | 1.09 (0.54, 2.23) | 243 | 244 | 75.4 | 0.864 |
| | Non-interactive | 4 | 1.04 (0.77, 1.39) | 1241 | 800 | 61.4 | |
| | Used mobile app | 4 | 1.23 (0.79, 1.92) | 371 | 372 | 71.5 | 0.442 |
| | No mobile app | 4 | 0.94 (0.54, 1.62) | 1113 | 672 | 65.3 | |
| | High overall risk of bias | 2 | 0.87 (0.15, 5.16) | 167 | 167 | 91.6 | 0.686 |
| | Low/some concerns overall risk of bias | 6 | 1.09 (0.86, 1.38) | 1371 | 877 | 45.8 | |
| | *Total* | **8** | **1.09 (0.81, 1.48)** | **1484** | **1044** | **66.9** | **–** |
| Pre-eclampsia | *Total* | **3** | **0.82 (0.51, 1.33)** | **979** | **528** | **0.0** | **–** |
| Preterm birth | *Total* | **2** | **0.83 (0.53, 1.28)** | **929** | **477** | **0.0** | **–** |

CI: confidence interval; BMI: body mass index.

**Cesarean birth (CB).** Eight trials [20,22,29,30,32,33,37,39] (2,528 participants) provided data on the rate of CB. We found a RR of 1.09 (95% CI 0.81 to 1.48; Table 3 and Figure M in S3 Appendix, moderate certainty) with an absolute effect of 2% more CB (from 5 fewer to 12 more) when pregnant individuals were provided with digital health intervention (Table 4) compared with routine care. Subgroup analysis found that the risk of CB may be slightly higher in women with higher BMI (BMI ≥ 25), but results overlapped across groups (Table 3 and Figure N in S3 Appendix).

**Table 4. GRADE summary of findings for mHealth interventions during pregnancy.**

| # of trials (# of patients) | Risk of bias | Inconsistency | Indirectness | Imprecision | Publication bias¹ | Relative effect (95% CI) | Absolute effect (95% CI) | Certainty of evidence |
|---|---|---|---|---|---|---|---|---|
| **Gestational weight gain (GWG) (BMI ≥ 25) – kilogram** | | | | | | | | |
| 8 (712) | Not serious | Not serious | Not serious | Serious* | Unde-tected | – | MD **−1.80** (−2.60 to −0.99) | Moderate |
| **Gestational weight gain (GWG) (any BMI) – kilogram** | | | | | | | | |
| 8 (2455) | Not serious | Not serious | Not serious | Not serious | Unde-tected | – | MD **−0.00** (−0.05 to 0.04) | High |
| **Gestational weight gain (GWG) exceeding Institute of Medicine (IOM) recommendations** | | | | | | | | |
| 14 (1939) | Not serious | Serious† | Not serious | Not serious | Unde-tected | **RR 0.80** (0.68 to 0.95) | **9 fewer per 100** (from 15 fewer to 2 fewer) | Moderate |
| **Postpartum weight retention** | | | | | | | | |
| 4 (217) | Measured at different time points and not pooled. ¾ studies found reductions at different time points. | | | | | | | |
| **Gestational diabetes mellitus (GDM)** | | | | | | | | |
| 7 (2262) | Not serious | Not serious | Not serious | Very serious‡ | Unde-tected | **RR 0.80** (0.57 to 1.12) | **3 fewer per 100** (from 6 fewer to 2 more) | Low |
| **Cesarean birth (CB)** | | | | | | | | |
| 8 (2528) | Not serious | Serious§ | Not serious | Not serious | Unde-tected | **RR 1.09** (0.81 to 1.48) | **2 more per 100** (from 5 fewer to 12 more) | moderate |
| **Pre-eclampsia** | | | | | | | | |
| 3 (1507) | Not serious | Not serious | Not serious | Very serious¶ | Unde-tected | **RR 0.82** (0.51 to 1.33) | **1 fewer per 100** (from 3 fewer to 2 more) | Low |
| **Preterm birth** | | | | | | | | |
| 2 (1354) | Not serious | Not serious | Not serious | Very serious# | Unde-tected | **RR 0.83** (0.53 to 1.28) | **1 fewer per 100** (from 3 fewer to 2 more) | Low |
| **Birthweight – kilogram** | | | | | | | | |
| 10 (2366) | Not serious | Not serious | Not serious | Not serious | Unde-tected | – | MD **0** (−0.08 to 0.08) | High |

**CI:** confidence interval; **MD:** mean difference; **RR:** risk ratio

1. Funnel plots are presented in Figure A and B S4 Appendix

**Explanations**

*The total sample size is lower than the optimal information size.

†There is high statistical heterogeneity, and the effect estimates of some studies are importantly different from each other.

‡The number of events (GDM) in this meta-analysis is only 232, which does not meet the optimal information size. Also, the upper bound of the 95% CI crosses the threshold of null effect.

§There is high statistical heterogeneity, and the effect estimates of some studies are importantly different from each other. which may explain the wide confidence interval. Therefore, the authors rated down 1 level due to inconsistency.

¶The number of events in this meta-analysis is 67, which does not meet the optimal information size. Also, the 95% CI crosses the line of no effect. Therefore, we rated down 2 levels owing to imprecision.

#Relative risk CI is wide likely due to few events and some heterogeneity, but CI around absolute effects is narrow. We therefore rated down twice for imprecision.

Analyses by the risk of bias of studies, and intervention type did not explain the differences in effects (Table 3 and Figures O-Q in S3 Appendix).

**Pre-eclampsia.** Three trials [20,30,37] (1,507 participants) provided data on the number of women with pre-eclampsia. We found the RR = 0.82 (95% CI: 0.51 to 1.33; low certainty Table 3 and Figure R in S3 Appendix) with an absolute effect of 1% fewer cases of pre-eclampsia (from 3% fewer to 2% more) in pregnant individual when provided with digital health intervention compared with routine care (Table 4).

Table 5. Summary of the results for studies reporting postpartum weight retention.

| Author (Year) | Follow up time | Postpartum weight measurement | Intervention Mean (SD) N | Comparator Mean (SD) N | P-value |
|---|---|---|---|---|---|
| Smith et. al (2014) | 4 weeks | Maternal weight retention was calculated by subtracting the woman's pre-pregnancy weight from her weight measured at the 1-month postpartum visit. | 5.3 (5.70) 24 | 3.9 (5.4) 21 | 0.67 |
| Rani et. al (2022) | 8 weeks | Post-partum weight retention was obtained by subtracting the pre-pregnancy weight from the weight measured at two months postdelivery | 5.6 (3.14) 59 | 8.62 (3.08) 29 | 0.0001 |
| Coughlin et. al (2020) | 12 weeks | Change in weight from 11–16 weeks gestation to 12 weeks after childbirth | 0.9 (4.5) 13 | 2.6 (3.7) 13 | 0.30 |
| Herring et. al (2017) | 24 Weeks | Subtracting Early pregnancy weight (<20 weeks) from weight at 24 weeks postpartum | 0.8 (14.3) 27 | 4.8 (13.8) 31 | 0.28 |

## Neonatal outcomes

**Birthweight.** Ten trials [20,22,24,25,29,31,33–35,37,39] (2,366 participants) provided data on birthweight. High certainty evidence suggested that weight management digital health interventions, compared with routine care, have no effect on birthweight in newborns (MD = 0.00; 95% CI: −0.08 to 0.08; Tables 2 and 4, Figure S in S3 Appendix). We did not perform subgroup analysis since the high statistical heterogeneity may be influenced by minor differences in weight that are not clinically significant, such as a 230g decrease (Chen study) [31] versus a 200g increase (Rani study) [35].

**Preterm birth.** Two studies [20,30] (1,402 participants) reported on the number of preterm births. We found a RR of 0.83 (95% CI: 0.53 to 1.28; low certainty) with an absolute effect of 1% fewer preterm birth cases (from 3 fewer to 2 more) in weight management digital health interventions, compared with routine care, (Tables 3 and 4 and Figure T in S3 Appendix).

## Discussion

In pregnant individuals with a singleton pregnancy, digital health weight management interventions have little to no effect on birthweight and GWG in subgroups including all BMI categories (high-certainty evidence). Digital health interventions probably reduce GWG in individuals with a BMI ≥ 25 (overweight or obesity), and probably reduce the risk of excessive GWG according to IOM recommendations and probably result in little to no effect in the rate of CB (Moderate-certainty evidence). Digital health weight management interventions may reduce GDM, may reduce pre-eclampsia, and may reduce preterm birth (Low-certainty evidence).

With regards to GWG, in the subgroup analysis based on the BMI categories, the results showed a higher reduction in GWG in studies that included only participants with BMI ≥ 25 kg/m². In contrast, in studies that recruited participants across all BMI categories, no such effect was found. This difference among BMI subgroups might indicate that the digital health interventions are more effective among higher BMI categories. However, more studies are needed to confirm these findings.

A previous systematic review on exclusively digital interventions for GWG management also reported a reduction in GWG in the intervention group compared with the control group; however, this reduction was reported as statistically non-significant [10]. That review included only six studies with separate analyses based on intention-to-treat (three studies, MD: −0.28 kg; 95% CI −1.43 to 0.87) and per-protocol (four studies, MD: −0.65 kg; 95% CI −1.98 to 0.67) [10], compared with our 18 studies, reflecting substantial growth in this field.

Sherifali et al. conducted a systematic review of eHealth technologies on weight management in pregnancy and the postpartum period, which differed from ours in terms of inclusion criteria and population [11]. Sherifali et al.'s review showed a reduction of 1.62 kg (95% CI −3.57, 0.33) in weight gain for pregnant women, compared with our overall finding of 0.78 kg 95% CI −1.40, −0.16) Their review included studies with participants with diabetes and some studies with

in-person components, whereas our review focused exclusively on digital health interventions among participants without pre-existing conditions, which may explain the higher reduction in their study compared to ours. However, the certainty of evidence for these two aforementioned systematic reviews was limited by the heterogeneity of the studies and the lack of precision in the results.

Another systematic review of pregnant women who were overweight or with obesity based on BMI showed that the mHealth intervention group gained 1.12 kg (95% CI −1.44, −0.80) less weight compared with the control group [40]. Our subgroup analysis result on overweight/obesity BMI categories showed a lower weight gain (−1.80 kg). Comparing our results directly to that review might not be appropriate for two reasons. First, our study included a wider range of participants, not just those with higher BMI categories. Second, the interventions were not exclusively digital-based; they also had in-person components.

A large systematic review (55 trials, 20,090 pregnancies) without restriction on the delivery type of the intervention (digital or in-person), found that counseling and behavioral interventions helped individuals gain less weight during pregnancy (1.02 kg less on average) [41]. However, the results varied widely across studies ($I^2 = 43\%$). Consistent with our study, they found a pattern indicating that the intervention group had lower mean GWG, as BMI increased. The biggest reduction (1.63 kg) was seen in individuals with obesity. These findings can be due to more rigorous monitoring and counseling regarding weight gain during pregnancy in individuals with higher BMI due to the higher risks associated with obesity in pregnancy [42].

With regards to GDM, while acknowledging methodological differences, our estimate (RR = 0.80) was aligned with other previous systematic reviews. He et al. [40] reported an OR=0.74 (95% CI 0.56,0.96) from 16 studies of mHealth interventions in pregnant women with overweight or obesity. Furthermore, He et al. [40] also reported a reduction in preterm birth using mHealth intervention (OR 0.65; 95% CI 0.48, 0.87), which was consistent with our study. However, their study was limited to pregnant women with overweight or obesity and some included studies that had in-person components.

Regarding CB, while we found little to no effect of digital health interventions, Leblalta et al. [43] reported a reduction in cesarean delivery rates (RR = 0.81; 95% CI: 0.69 to 0.95; high certainty evidence) in their systematic review of digital health interventions. The contrasting findings likely reflect important differences in study populations and intervention design, with their review focusing exclusively on women with GDM who may derive greater benefit from intensive digital monitoring and glucose management, whereas our review included pregnant individuals across all risk categories for general weight management. Additionally, their review included studies with in-person components alongside digital interventions, whereas our review focused exclusively on digital-only interventions.

Our subgroup analysis indicated that interactive interventions and mobile applications may be more effective in reducing GWG and the risk of excessive GWG compared to non-interactive interventions and other types, such as text messages or websites. However, these findings require cautious interpretation due to overlapping confidence intervals between subgroups, and smaller number of studies in the interactive and mobile application subgroups, which collectively limit our ability to draw definitive conclusions about the superiority of specific intervention types. Most included studies employed multi-component approaches combining various behavior change techniques such as goal-setting, self-monitoring, educational content delivery, and personalized feedback, delivered through diverse platforms including text messaging, mobile applications, websites, and group-based interactions, making it challenging to isolate which specific intervention components drive effectiveness. The heterogeneity in intervention design, delivery methods, and behavioral targets across studies prevented us from conducting more detailed subgroup analyses to identify the most effective intervention elements. Nevertheless, these preliminary results suggest that future research can prioritize interactive and mobile application-based interventions while systematically evaluating individual intervention components to determine the optimal combination of features for gestational weight management.

The limited number of studies reporting certain outcomes, particularly preterm birth (2 studies) and pre-eclampsia (3 studies), necessitates cautious interpretation of these findings. While our results suggest potential benefits, low certainty

of evidence due to wide confidence intervals, and low number of events indicate that these effects require confirmation in larger, adequately powered studies before clinical recommendations can be made.

## Strengths and Limitations

Strengths of this review include the rigorous systematic review methodology following Cochrane handbook for systematic reviews and meta-analysis, and a comprehensive search that identified several trials not considered in prior reviews. Further, we considered patient-important outcomes related to both pregnant individuals and children, explored *a priori* identified subgroup effects, and used the GRADE approach to rate the certainty of evidence.

The evidence we found has limitations. Some studies used self-reported pre-pregnancy and delivery weights to assess total GWG, which might introduce a systematic bias in estimating GWG. However, we did not rate down the certainty of evidence since this bias was not specific to a group in the studies (non-differential misclassification). According to a systematic review, women tend to underreport their pre-pregnancy and delivery weights, though the magnitude of error was relatively small and correlation between self-reported pre-pregnancy weight and measured weights remained high [44]. Although it seems most likely that any measurement error would be non-differential, we cannot rule out the possibility of differential measurement error that may bias results either towards or away from the null. Future RCTs should therefore prioritize objectively measured weights when feasible Secondly, all the studies except for one were conducted in high-income countries. This can affect the generalizability of our results to low- and middle-income countries (LMICs), where smartphone and internet penetration may be lower, healthcare infrastructure differs, cultural attitudes toward technology-mediated healthcare may vary, or pregnancy-related outcomes and risk factors have different baselines. Future research should assess the impact of digital health interventions in diverse healthcare systems. Thirdly, the type of digital health intervention varied among the studies, which limits our ability to identify the most effective intervention characteristics. Although we explored this heterogeneity by performing two subgroup analyses (mobile application based and interactive based) as data permitted. The future research should investigate the most effective components of digital health interventions during pregnancy, such as frequency of contact, content and timing of messages, types of behavioral change techniques employed, intervention duration, and level of personalization. Rigorous studies with more comparable interventions and larger participant numbers are needed to draw definitive conclusions, especially concerning different BMI categories and the effectiveness of various types of digital health interventions, including their interactivity and use of mobile applications.

## Conclusions

Our systematic review explored the efficacy of weight management digital health interventions in pregnant people versus routine care. Our review showed that digital health interventions are likely to reduce GWG in individuals with a BMI ≥ 25 kg/m$^2$ (moderate-certainty evidence) and also the risk of excessive GWG according to IOM guidelines (moderate-certainty evidence). These interventions have little to no effect on GWG across the studies included all BMI categories (high-certainty evidence) and little to no difference in CB rates (moderate-certainty evidence). Digital health interventions may reduce the risk of GDM, pre-eclampsia, and preterm birth (low-certainty evidence) and have little to no effect on birthweight (high-certainty evidence). These findings underscore the potential of digital health interventions to improve weight management and mitigate associated risks during pregnancy, especially for those with overweight or obesity. However, more rigorous studies with larger sample sizes and comparable methodologies are needed. Future research should also focus on identifying the most effective components of digital health interventions and ensuring their applicability across diverse populations.

## Supporting information

**S1 Appendix. Search strategy for MEDLINE, Embase, PsychInfo, and Proquest.**
(DOCX)

**S2 Appendix. Risk of bias assessment.**
(DOCX)

**S3 Appendix. Forest plots for meta-analyses.**
(DOCX)

**S4 Appendix. Funnel plots.**
(DOCX)

**S1 Table. Characteristics of excluded studies and reasons for exclusion.**
(DOCX)

## Author contributions

**Conceptualization:** Sahar Khademioore, Ahmad Sofi-Mahmudi, Kiara Pannozzo, Rohan D'Souza, Gian Paolo Morgano, Nancy Santesso, Laura N. Anderson.

**Data curation:** Sahar Khademioore, Alexandra M. Palumbo, Taylor Incze, Nicolette Christodoulakis.

**Formal analysis:** Sahar Khademioore, Ahmad Sofi-Mahmudi, Gian Paolo Morgano.

**Investigation:** Sahar Khademioore, Taylor Incze, Kiara Pannozzo, Gian Paolo Morgano, Nancy Santesso, Laura N. Anderson.

**Methodology:** Sahar Khademioore, Alexandra M. Palumbo, Ahmad Sofi-Mahmudi, Gian Paolo Morgano, Nancy Santesso, Laura N. Anderson.

**Project administration:** Sahar Khademioore.

**Supervision:** Sahar Khademioore, Rohan D'Souza, Gian Paolo Morgano, Nancy Santesso, Laura N. Anderson.

**Validation:** Sahar Khademioore, Alexandra M. Palumbo, Ahmad Sofi-Mahmudi, Nicolette Christodoulakis, Rohan D'Souza, Gian Paolo Morgano, Nancy Santesso, Laura N. Anderson.

**Visualization:** Sahar Khademioore, Ahmad Sofi-Mahmudi, Kiara Pannozzo, Nicolette Christodoulakis, Nancy Santesso.

**Writing – original draft:** Sahar Khademioore.

**Writing – review & editing:** Sahar Khademioore, Alexandra M. Palumbo, Ahmad Sofi-Mahmudi, Taylor Incze, Kiara Pannozzo, Nicolette Christodoulakis, Rohan D'Souza, Gian Paolo Morgano, Nancy Santesso, Laura N. Anderson.

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
