## [Decision Letter · Decision Letter 0]

15 Jul 2025

PONE-D-25-01046Effects of mobile health counseling and behavioral interventions on weight management during pregnancy and postpartum: A systematic review and meta-analysis of randomized controlled trialsPLOS ONE

Dear Dr. Khademioore,

Thank you for submitting your manuscript to PLOS ONE. After careful consideration, we feel that it has merit but does not fully meet PLOS ONE’s publication criteria as it currently stands. Therefore, we invite you to submit a revised version of the manuscript that addresses the points raised during the review process.

We look forward to receiving your revised manuscript.

Kind regards,

Dr Anh Nguyen

Academic Editor

PLOS ONE

Journal Requirements:

**Additional Editor Comments:**

Overall, the paper provides a valuable review of technology-based interventions for weight management during pregnancy and postpartum. Some areas require clarification and improvement:

• Please provide a clear definition of "mHealth" in the introduction, as the term covers a wide range of interventions. Given the diverse delivery modes in your included studies (e.g., mobile apps, websites, text messaging, Zoom), a more specific and consistent definition is needed. Additionally, the current title "mobile health counseling and behavioral interventions" does not fully reflect the scope of the review. Consider revising it to "Digital health interventions"

• The review presents two subgroup analyses of intervention types; however, the differences are not statistically significant. If possible, please provide a more detailed and cautious interpretation of these findings

• In the discussion (lines 286–301), direct comparisons with previous systematic reviews are not appropriate due to major differences in inclusion criteria, populations, and interventions. These differences should be clearly acknowledged and carefully contextualized.

• Please clarify whether only English-language studies were included. If so, this should be noted as a limitation due to potential language bias.

Reviewers' comments:

Reviewer's Responses to Questions

**Comments to the Author**

1. Is the manuscript technically sound, and do the data support the conclusions?

Reviewer #1: Partly

Reviewer #2: Partly

2. Has the statistical analysis been performed appropriately and rigorously? 

Reviewer #1: Yes

Reviewer #2: Yes

3. Have the authors made all data underlying the findings in their manuscript fully available?

Reviewer #1: Yes

Reviewer #2: Yes

4. Is the manuscript presented in an intelligible fashion and written in standard English?

Reviewer #1: Yes

Reviewer #2: Yes

5. Review Comments to the Author

Reviewer #1: This article systematically reviews and meta-analyzes randomized controlled trials (RCTs) evaluating mHealth (mobile health) interventions, ranging from apps and text messages to online platforms, for weight management during pregnancy and postpartum. The authors aimed to determine whether these digital behavioral and counseling interventions reduce gestational weight gain (GWG), excessive GWG, and improve other maternal and neonatal outcomes. The review included 18 RCTs with varying sample sizes and populations, with a notable subgroup finding that women with a BMI ≥ 25 kg/m² experienced a statistically significant reduction in GWG compared with routine care. Overall, the evidence was graded from high to low certainty depending on the outcome.

The included studies appear generally adequate to address the research question, although the reliance on self-reported pre-pregnancy weight in several studies is a limitation that may introduce nondifferential misclassification of the primary outcome.

The meta-analytic approach is appropriate for the research questions posed. Strengths of the statistical approach include clear reporting of effect sizes, confidence intervals, and tests for interactions. However, some outcomes (e.g., preterm birth and pre-eclampsia) were based on only a few studies, resulting in wide confidence intervals and a lower level of certainty. This highlights a key weakness: while the overall methodology is statistically sound, the imprecision in certain subgroup analyses and the inherent heterogeneity in intervention designs limit the strength of some conclusions.

The diversity of mHealth interventions (from text messages to interactive mobile apps) makes it difficult to determine which components drive effectiveness. Although subgroup analyses were conducted, further sensitivity analyses could help clarify the robustness of the results.

Outcomes such as preterm birth and pre-eclampsia are based on few studies, resulting in wide confidence intervals and low certainty of evidence.

Future iterations or updates could benefit from a more detailed breakdown of which aspects of mHealth interventions (e.g. frequency of contact, content of messages) are most effective.

Many studies used self-reported pre-pregnancy weight, which could introduce bias. Although the authors noted the use of self-reported weights, a more thorough discussion of their potential impact on the overall results would be useful.

As almost all of the studies were conducted in developed countries, the findings may not be generalizable to settings with fewer resources. Additional commentary on the implications of conducting the majority of studies in high-income settings and suggestions for future research in more diverse populations would strengthen the discussion.

I did not notice any significant spelling errors, but the word "counseling" is used several times. You may want to consider replacing it with "counseling".

Reviewer #2: An interesting and well written systematic review paper. A few suggestions as to how to strengthen the paper for readers of this journal

1) Please clarify the aim of the review throughout the manuscript. The abstract states ‘This systematic review evaluated the effects of counselling or behavioral mHealth interventions to prevent excessive GWG’, however the objective refers ‘This systematic review aimed to evaluate the effects of mHealth counselling or behavioral weight management interventions among pregnant individuals of all BMI categories, compared to routine care or in-person delivery of interventions’ This will help readers understand the context

2) Introduction could be more focused to provide a clear rationale regarding the types of interventions and set the scene for the review? Is this about healthy eating, activity or is it more about behavioural techniques. A broad summary of the types on interventions would be useful to give focus. Additionally the introduction needs to clearly explain if this review is focusing on weight loss as an outcome or is it other outcomes which are of interest

3) Methods an explanation of ‘usual care’ and ‘mHealth’. A clear definition as used by authors for searching is required

4) Methods: include years included to explain how old some studies are?

5) Methods: include how GWG was calculated in the selected studies generally and refer to limitations regarding self reported.

5) Discussion is unclear in places and could be more clearly explained throughout for readers. For example Moderate-certainty evidence suggests that mHealth interventions probably reduce GWG in individuals with a BMI≥25 ? please add more information to explain ‘probably reduce GWG’

Low-certainty evidence suggests that mHealth weight management interventions may reduce the risk of GDM, pre eclampsia, and preterm birth. Please clarify what this means with regards reduce the risk of GDM by adding values/numbers

Limitations: more information is needed here to justify the conclusions. It states that pre pregnancy weight was self reported. There is no comments about the last weight recorded in pregnancy? Therefore these limitations must be acknowledged within the discussion to help put the results in context

6) It would be useful to readers to have more detail about the type of interventions which are more ‘successful’ this would add more depth to the discussion

6) Overall please clearly outline the findings from this review and provide a key message(s) for readers; Could this be more clearly and specifically stated for example ‘Our review showed that mHealth interventions are likely to reduce GWG in individuals with a BMI ≥ 25 kg/m2 and also the risk of excessive GWG according to IOM guidelines’. Does this mean that m Health are useful?

6. PLOS authors have the option to publish the peer review history of their article (what does this mean? ). If published, this will include your full peer review and any attached files.

**Do you want your identity to be public for this peer review?** For information about this choice, including consent withdrawal, please see our Privacy Policy .

Reviewer #1: No

Reviewer #2: No

---

## [Author Response · Author response to Decision Letter 1]

1 Aug 2025

EDITOR COMMENTS:

Overall, the paper provides a valuable review of technology-based interventions for weight management during pregnancy and postpartum. Some areas require clarification and improvement:

Comment 1: Please provide a clear definition of "mHealth" in the introduction, as the term covers a wide range of interventions. Given the diverse delivery modes in your included studies (e.g., mobile apps, websites, text messaging, Zoom), a more specific and consistent definition is needed. Additionally, the current title "mobile health counseling and behavioral interventions" does not fully reflect the scope of the review. Consider revising it to "Digital health interventions"

Response: Thank you for highlighting the need for a more accurate terminology. We agree that our systematic review covers a broad range of digital interventions, and we have changed our title to "digital health" to better reflect our objective, and have also used the term "digital health" instead of "mHealth" throughout the manuscript.

We have also added a comprehensive definition of digital health to our Introduction as follows:

Lines 86-89: "Digital health encompasses the use of information and communication technologies to improve healthcare delivery and patient outcomes.6 This field includes mobile health applications, web-based platforms, telemedicine, and health informatics systems. Digital health interventions are increasingly used to support health behavior change and healthcare delivery, offering opportunities to reach individuals outside traditional clinical settings."

Additionally, we have defined the specific interventions eligible for our study in the Methods section Lines 134-136: "For this review, 'digital health' was defined as interventions which were delivered primarily through mobile devices, tablets, or computers, including mobile applications, text messaging, websites, email communications, and online platforms accessible via mobile devices."

Comment 2: The review presents two subgroup analyses of intervention types; however, the differences are not statistically significant. If possible, please provide a more detailed and cautious interpretation of these findings.

Response: Thank you for raising this important point. We have revised our interpretation of subgroup analyses to be more cautious, while reflecting the higher effect sizes in the interactive and app-based groups. We now clearly acknowledge the limitations and provide more detailed interpretation in Results and Discussion section:

Results Lines 237-245: " There was no evidence of a statistically significant interaction in subgroup analyses by the risk of bias and intervention type (interactive vs. non-interactive and mobile application vs. other delivery methods) in exploring the heterogeneity (interaction tests p > 0.05, Table 3 and Supplementary Figures 9–12). Although these interactions were not statistically significant and confidence intervals overlapped, pooled effect estimates suggest that interactive interventions (-1.51 kg; 95% CI: -2.71 to -0.32 vs. -0.48 kg; 95% CI: -1.14 to 0.19) and interventions mobile applications (-1.06 kg; 95% CI: -1.85 to -0.27 vs. -0.55 kg; 95% CI: -1.54 to 0.44) may reduce GWG more effectively than non-interactive interventions and non-mobile application interventions."

Discussion section Lines 362-365: “However, these findings require cautious interpretation due to overlapping confidence intervals between subgroups, and smaller number of studies in the interactive and mobile application subgroups, which collectively limit our ability to draw definitive conclusions about the superiority of specific intervention types.”

Comment 3: In the discussion (lines 286–301), direct comparisons with previous systematic reviews are not appropriate due to major differences in inclusion criteria, populations, and interventions. These differences should be clearly acknowledged and carefully contextualized.

Response: We have thoroughly revised the Discussion to acknowledge differences and provide proper contextualization:

Lines 319-326: " Sherifali et al. conducted a systematic review of eHealth technologies on weight management in pregnancy and the postpartum period, which differed from ours in terms of inclusion criteria and population.11 Sherifali et al.’s review showed a reduction of 1.62 kg (95% CI -3.57, 0.33) in weight gain for pregnant women, compared to our overall finding of 0.78 kg 95% CI -1.40, -0.16). Their review included studies with participants with diabetes and some studies with in-person components, whereas our review focused exclusively on digital health interventions among participants without pre-existing conditions, which may explain the higher reduction in their study compared to ours. "

Comment 4: Please clarify whether only English-language studies were included. If so, this should be noted as a limitation due to potential language bias.

Response: In Line 145 of our Methods section, we specified that there were no language restrictions in our search to limit potential language bias. However, all papers that were included and eligible, although from non-English speaking countries, were published in English. We have added this information to our Results section:

Line 204: "All studies were published in English, between 2014 and 2024."

REVIEWER #1 COMMENTS:

Comment 1: This article systematically reviews and meta-analyzes randomized controlled trials (RCTs) evaluating mHealth (mobile health) interventions, ranging from apps and text messages to online platforms, for weight management during pregnancy and postpartum. The authors aimed to determine whether these digital behavioral and counseling interventions reduce gestational weight gain (GWG), excessive GWG, and improve other maternal and neonatal outcomes. The review included 18 RCTs with varying sample sizes and populations, with a notable subgroup finding that women with a BMI ≥ 25 kg/m² experienced a statistically significant reduction in GWG compared with routine care. Overall, the evidence was graded from high to low certainty depending on the outcome.

The included studies appear generally adequate to address the research question, although the reliance on self-reported pre-pregnancy weight in several studies is a limitation that may introduce nondifferential misclassification of the primary outcome.

Response: Thank you for summarizing our results so clearly and for highlighting this limitation. Yes, some studies used self-reported pre-pregnancy weight in the calculation of gestational weight gain. We have now expanded on the effect of self-reported weight in our Limitations section:

Lines 389-398: " Some studies used self-reported pre-pregnancy and delivery weights to assess total GWG, which might introduce a systematic bias in estimating GWG. However, we did not rate down the certainty of evidence since this bias was not specific to a group in the studies (non-differential misclassification). According to a systematic review, women tend to underreport their pre-pregnancy and delivery weights, though the magnitude of error was relatively small and correlation between self-reported pre-pregnancy weight and measured weights remained high.44 Although it seems most likely that any measurement error would be non-differential, we cannot rule out the possibility of differential measurement error that may bias results either towards or away from the null. Future trials should therefore prioritize objectively measured weights when feasible."

Comment 2: The meta-analytic approach is appropriate for the research questions posed. Strengths of the statistical approach include clear reporting of effect sizes, confidence intervals, and tests for interactions. However, some outcomes (e.g., preterm birth and pre-eclampsia) were based on only a few studies, resulting in wide confidence intervals and a lower level of certainty. This highlights a key weakness: while the overall methodology is statistically sound, the imprecision in certain subgroup analyses and the inherent heterogeneity in intervention designs limit the strength of some conclusions.

Response: Thank you for these encouraging comments about our methodological strengths. We fully agree with the limitations imposed by insufficient data for certain outcomes. We have addressed these concerns in our GRADE assessments (Table 2) as we appropriately downgraded certainty of evidence for imprecision when confidence intervals were wide or when few events occurred, resulting in low certainty evidence for outcomes such as pre-eclampsia and preterm birth. Additionally, in Results section, we explicitly noted the number of contributing studies for all outcomes and number of events. We have now also added a note on these limitations in the Discussion section.

Lines 377-381: "The limited number of studies reporting certain outcomes, particularly preterm birth (2 studies) and pre-eclampsia (3 studies), necessitates cautious interpretation of these findings. While our results suggest potential benefits, low certainty of evidence due to wide confidence intervals, and low number of events indicate that these effects require confirmation in larger, adequately powered studies before clinical recommendations can be made."

Comment 3: The diversity of mHealth interventions (from text messages to interactive mobile apps) makes it difficult to determine which components drive effectiveness. Although subgroup analyses were conducted, further sensitivity analyses could help clarify the robustness of the results.

Response: Thank you for bringing up this very important point. We agree that identifying the most effective components in mHealth interventions is necessary for future application of these tools in clinical practice. By subgrouping studies that used an application or had an interactive intervention with a health provider, we tried to delve deeper into this matter; however, the data did not allow us to perform more analyses as we also explain in respond to Comment 5. We have added this to the Discussion:

Lines 366-372: "Most included studies employed multi-component approaches combining various behavior change techniques such as goal-setting, self-monitoring, educational content delivery, and personalized feedback, delivered through diverse platforms including text messaging, mobile applications, websites, and group-based interactions, making it challenging to isolate which specific intervention components drive effectiveness. The heterogeneity in intervention design, delivery methods, and behavioral targets across studies prevented us from conducting more detailed subgroup analyses to identify the most effective intervention elements."

Comment 4: Outcomes such as preterm birth and pre-eclampsia are based on few studies, resulting in wide confidence intervals and low certainty of evidence.

Response We have addressed this concern in Discussion as explained in our previous response to Comment 2.

Comment 5: Future iterations or updates could benefit from a more detailed breakdown of which aspects of mHealth interventions (e.g. frequency of contact, content of messages) are most effective.

Response: We attempted to address this by examining interactive vs. non-interactive interventions and mobile app vs. non-mobile app interventions, as permitted by available data. However, further exploration of specific components (such as frequency of contact, content of messages, or behavioral change techniques) could not be conducted, as most studies employed multi-component interventions and they did not report outcomes stratified by individual intervention components

We have addressed this limitation in our Discussion section as mentioned in our previous response to Comment 3.

Comment 6: Many studies used self-reported pre-pregnancy weight, which could introduce bias. Although the authors noted the use of self-reported weights, a more thorough discussion of their potential impact on the overall results would be useful.

Response: Thank you for this feedback. We have now expanded our Limitations section to clearly address this issue as detailed in our response above to comment 1 (Lines 389-398).

Comment 7: As almost all of the studies were conducted in developed countries, the findings may not be generalizable to settings with fewer resources. Additional commentary on the implications of conducting the majority of studies in high-income settings and suggestions for future research in more diverse populations would strengthen the discussion.

Response: Thank you for this insightful comment regarding the generalizability of our findings. We have expanded the Discussion section to additionally comment on this point:

Lines 398-403: "All the studies except for one were conducted in high-income countries. This can affect the generalizability of our results to low- and middle-income countries (LMICs), where smartphone and internet penetration may be lower, healthcare infrastructure differs, cultural attitudes toward technology-mediated healthcare may vary, or pregnancy-related outcomes and risk factors have different baselines. Future research should assess the impact of digital health interventions in diverse healthcare systems."

Comment 8: I did not notice any significant spelling errors, but the word "counseling" is used several times. You may want to consider replacing it with "counseling".

Response: Thank you for catching this inconsistency. We have standardized the spelling throughout the manuscript using American English conventions ("counseling") consistent with the journal's requirements.

REVIEWER #2 COMMENTS:

Comment 1: An interesting and well written systematic review paper.

Response: Thank you.

Comment 2: Please clarify the aim of the review throughout the manuscript. The abstract states 'This systematic review evaluated the effects of counselling or behavioral mHealth interventions to prevent excessive GWG', however the objective refers 'This systematic review aimed to evaluate the effects of mHealth counselling or behavioral weight management interventions among pregnant individuals of all BMI categories, compared to routine care or in-person delivery of interventions' This will help readers understand the context.

Response: We appreciate this feedback regarding consistency. We have revised the manuscript to ensure consistent objectives throughout:

Abstract (Lines 40-43): " This systematic review aimed to evaluate the effects of digital health counseling or behavioral weight management interventions for preventing excessive gestational weight gain (GWG) among pregnant individuals of all body mass index (BMI) categories, compared to routine care."

Objective section (Lines 114-116): "This systematic review aimed to evaluate the effects of mHealth counseling or behavioral weight management interventions among pregnant individuals of all BMI categories, compared to routine care or in-person delivery of interventions."

Comment 3: Introduction could be more focused to provide a clear rationale regarding the types of interventions and set the scene for the review? Is this about healthy eating, activity or is it more about behavioural techniques. A broad summary of the types on interventions would be useful to give focus. Additionally the introduction needs to clearly explain if this review is focusing on weight loss as an outcome or is it other outcomes which are of interest.

Response: Thank you for this valuable feedback. We have revised our Introduction to better convey the gap in evidence and focus of our review:

Lines 82-85: "These interventions typically employ multiple behavioral change techniques including dietary counseling, physical activity promotion, self-monitoring strategies, and educational components delivered through face-to-face sessions."

Lines 86-90: Digital health encompasses the use of information and communication technologies to improve healthcare delivery and patient outcomes.6 This field includes mobile health applications, web-based platforms, telemedicine, and health informatics systems. Digital health interventions are increasingly used to support health behavior change and healthcare delivery, offering opportunities to reach individuals outside tradit

---

## [Editor Report · Decision Letter 1]

24 Aug 2025

Effects of digital health counseling and behavioral interventions on weight management during pregnancy and postpartum: A systematic review and meta-analysis of randomized controlled trials

PONE-D-25-01046R1

Dear Dr. Khademioore,

We’re pleased to inform you that your manuscript has been judged scientifically suitable for publication and will be formally accepted for publication once it meets all outstanding technical requirements.

Kind regards,

Dr. Anh Nguyen

Academic Editor

PLOS ONE

---

## [Editor Report · Acceptance letter]

PONE-D-25-01046R1

PLOS ONE

Dear Dr. Khademioore,

I'm pleased to inform you that your manuscript has been deemed suitable for publication in PLOS ONE. Congratulations! Your manuscript is now being handed over to our production team.

Kind regards,

on behalf of

Dr. Anh Nguyen

Academic Editor

PLOS ONE